# A Lower CD4 Count Predicts Most Causes of Death except Cardiovascular Deaths. The Austrian HIV Cohort Study

**DOI:** 10.3390/ijerph182312532

**Published:** 2021-11-28

**Authors:** Gisela Leierer, Armin Rieger, Brigitte Schmied, Mario Sarcletti, Angela Öllinger, Elmar Wallner, Alexander Egle, Manfred Kanatschnig, Alexander Zoufaly, Michele Atzl, Michaela Rappold, Ziad El-Khatib, Bruno Ledergerber, Robert Zangerle

**Affiliations:** 1Department of Dermatology and Venereology, Medical University of Innsbruck, 6020 Innsbruck, Austria; gisela.leierer@tirol-kliniken.at (G.L.); mario.sarcletti@tirol-kliniken.at (M.S.); michaela.rappold@tirol-kliniken.at (M.R.); 2Austrian HIV Cohort Study, 6020 Innsbruck, Austria; 3Division of Immunology, Allergy and Infectious Diseases, Department of Dermatology, Medical University of Vienna, 1090 Vienna, Austria; armin.rieger@meduniwien.ac.at; 4Otto-Wagner Hospital, 1140 Vienna, Austria; brigitte.schmied@gesundheitsverbund.at; 5Department of Dermatology, Med. Campus III, Kepler University Hospital Linz, Johannes Kepler University Linz, 4040 Linz, Austria; angela.oellinger@kepleruniklinikum.at; 6Department of Internal Medicine, General Hospital Graz South-West, 8020 Graz, Austria; elmar.wallner@kages.at; 7Department of Internal Medicine III with Hematology, Medical Oncology, Hemostaseology, Infectious Diseases, Rheumatology, Oncologic Center, Laboratory for Immunological and Molecular Cancer Research, Paracelsus Medical University, 5020 Salzburg, Austria; a.egle@salk.at; 81st Medical Department, General Hospital Klagenfurt, 9020 Klagenfurt, Austria; manfred.kanatschnig@kabeg.at; 9Kaiser-Franz-Josef Hospital Vienna, 1100 Vienna, Austria; alexander.zoufaly@wienkav.at; 10Department of Internal Medicine, Oncology, Haematology, Infectious Diseases, State Hospital Feldkirch, 6800 Feldkirch, Austria; michele.atzl@lkhf.at; 11Department of Surveillance and Infectious Disease Epidemiology, Austrian Agency for Health and Food Safety (AGES), Austrian Ministry for Health, Ministry of Sustainability and Tourism, 1220 Vienna, Austria; ziad.el-khatib@ages.at; 12Department of Social and Preventive Medicine, Medical University of Vienna, 1090 Vienna, Austria; 13Division of Infectious Diseases and Hospital Epidemiology, University Hospital Zurich, University of Zurich, 8006 Zurich, Switzerland; infled@usz.uzh.ch

**Keywords:** cohort study, cause-specific mortality, CD4, cART, risk factors

## Abstract

(1) Objective: To investigate changes in mortality rates and predictors of all-cause mortality as well as specific causes of death over time among HIV-positive individuals in the combination antiretroviral therapy (cART) era. (2) Methods: We analyzed all-cause as well as cause-specific mortality among the Austrian HIV Cohort Study between 1997 and 2014. Observation time was divided into five periods: Period 1: 1997–2000; period 2: 2001–2004; period 3: 2005–2008; period 4: 2009–2011; and period 5: 2012–2014. Mortality rates are presented as deaths per 100 person-years (d/100py). Potential risk factors associated with all-cause mortality and specific causes of death were identified by using multivariable Cox proportional hazard models. Models were adjusted for time-updated CD4, age and cART, HIV transmission category, population size of residence area and country of birth. To assess potential nonlinear associations, we fitted all CD4 counts per patient using restricted cubic splines with truncation at 1000 cells/mm^3^. Vital status of patients was cross-checked with death registry data. (3) Results: Of 6848 patients (59,704 person-years of observation), 1192 died: 380 (31.9%) from AIDS-related diseases. All-cause mortality rates decreased continuously from 3.49 d/100py in period 1 to 1.40 d/100py in period 5. Death due to AIDS-related diseases, liver-related diseases and non-AIDS infections declined, whereas cardiovascular diseases as cause of death remained stable (0.27 d/100py in period 1, 0.10 d/100py in period 2, 0.16 d/100py in period 3, 0.09 d/100py in period 4 and 0.14 d/100py in period 5) and deaths due to non-AIDS-defining malignancies increased. Compared to latest CD4 counts of 500 cells/mm^3^, lower CD4 counts conferred a higher risk of deaths due to AIDS-related diseases, liver-related diseases, non-AIDS infections and non-AIDS-defining malignancies, whereas no significant association was observed for cardiovascular mortality. Results were similar in sensitivity analyses where observation time was divided into two periods: 1997–2004 and 2005–2014. (4) Conclusions: Since the introduction of cART, risk of death decreased and causes of death changed. We do not find evidence that HIV-positive individuals with a low CD4 count are more likely to die from cardiovascular diseases.

## 1. Introduction

Life expectancy of HIV-positive individuals has increased as mortality rates have markedly declined because of the widely used combination antiretroviral therapy (cART) [1,2,3,4]. The most potential prognostic factor for death in the cArt era was CD4 cell count, while viral load, independent of CD4 cell count, was hardly predictive of death [5,6]. CD4 cell count at the start of cART adds more information about a person’s clinical risk than current CD4 count after having started cART [7,8]. The latest absolute CD4 count, however, is more closely associated with mortality than the baseline CD4 count [9,10,11] or the slope of the CD4 T-cell increase [12]. Previous studies have evaluated the utility of CD4 cell count to predict all-cause mortality. However, with an increased duration of the cART era, numerous reports have described not only a decline on mortality rates but also a change in causes of death. While AIDS-related mortality remains an important cause of death, increasing numbers of death due to non-AIDS-defining conditions such as malignancies, infections, liver diseases and cardiovascular diseases were reported [13,14].

Therefore, studies investigated both HIV-related risk factors and non HIV-related factors and their associations with specific causes of death [13,15,16]. An association was found between non-AIDS death causes and a longer time spent on cART as well as higher CD4 count at initiation of cART [17]. A large cohort study reported lower CD4 counts at the start of ART to be associated with higher rates of deaths due to AIDS-defining infections, renal failure, non-AIDS malignancy and other causes but not due to AIDS malignancies, whereas higher CD4 counts at the start of cART were found to be associated with death due to violent causes [13]. In contrast to these association studies with CD4 counts at the onset of cART, there are limited data regarding time-updated CD4 counts and their associations with specific causes of death. Similarly, data are scarce when taking into account even all measured CD4 counts throughout disease history. Therefore, such potential associations with the risk for specific non-AIDS deaths are not well described.

This study examines the prognostic value of a time-updated CD4 cell count to predict specific causes of death.

## 2. Materials and Methods

### 2.1. The Austrian HIV Cohort Study

The Austrian HIV Cohort Study (AHIVCOS) [18,19] is an open, multicentre, prospective, observational cohort study of HIV-infected individuals followed at seven HIV treatment centres in Austria. The study was initiated in 2001 as an incorporated association by representatives of five Austrian HIV treatment centres (AKH Vienna, Otto-Wagner-Hospital Vienna, AKH Linz, LKH Innsbruck and LKH Graz West). In 2008, two further HIV treatment centres (LKH Salzburg and LKH Klagenfurt) joined AHIVCOS, thus patients are currently enrolled actively and prospectively at seven public hospital-based HIV treatment centres covering approximately 80% of all treated HIV-infected patients in Austria.

In January 2016, AHIVCOS included information on 8696 individuals. The data on the following indicators were collected: Demographics (age, gender, province), clinical history, laboratory results, treatment history (types, doses and changes in cART including reasons for interruptions) and any coinfections (e.g., Hepatitis B and C, and syphilis).

### 2.2. Selection of Patients for Analysis

We analysed data from 6848 patients who had a visit event at least once at one of the seven HIV treatment centres in Austria between 1 January 1997 and 31 December 2014. Patients were followed until death or 6 months after last patient contact or 31 December 2014 depending on the date which first occurred. All data and analyses described in this manuscript were based on these 6848 patients.

### 2.3. Classification of Causes of Death

Causes of death were determined according to the “Coding Causes of Death in HIV” (CoDe) [20] classification system. In our analyses the underlying cause of death which represents the morbidity that initiated the sequence of events leading directly or indirectly to death was considered. All deaths were reviewed centrally by one physician in order to ensure that they were equally handled when assigning the causes of death. The vital status of patients was cross-checked with death registry data. For certain analyses they were grouped into the following categories: AIDS related, liver related (including HCV with cirrhosis, HCV with liver failure, HCV with liver cancer, HBV with cirrhosis, HBV with liver failure, HBV with liver cancer), non-AIDS-defining malignancies, non-AIDS infections, cardiovascular.

### 2.4. Statistical Methods

Data were presented as mean ± SD and median (interquartile range (IQR)) or number (%). Categorical data were compared using χ^2^ test, continuous variables were analysed using the non-parametric Mann–Whitney U test. Mortality rates of five time periods (1997–2000, 2001–2004, 2005–2008, 2009–2011, 2012–2014) are reported as deaths per 100 person-years (d/100py) of follow-up with 95% confidence intervals (CIs) calculated using the normal approximation of the Poisson distribution. To investigate the influence of various demographic and clinical parameters on all-cause mortality as well as specific causes of death, multivariable Cox regression models were performed. Observation time was divided into five periods, for each of them a multivariable Cox regression model was calculated. Covariates considered were CD4, age and cART as time-updated covariates, HIV transmission category, population size of residence area and country of birth. Furthermore, for each specific cause of death a multivariable Cox regression model was calculated separately adjusting for the same covariates. Additional sensitivity analyses were run where observation time was divided into two periods: 1997–2004 and 2005–2014. To assess potential nonlinear associations, we fitted all CD4 counts per patient using restricted cubic splines with truncation at 1,000 cells/mm^3^. The proportional hazard assumptions were checked for by testing for zero slopes of scaled Schoenfeld residuals. All analyses were conducted using Stata software, version 13.1 (StataCorp, College Station, TX, USA).

## 3. Results

### 3.1. Patient Characteristics

Characteristics of 6848 patients contributing 59,706 person-years of observation with a median follow-up time of 7.78 years (IQR: 3.75–13.67) are described in Table 1.

Median age at study entry was 34.5 years and the majority of patients were male (74.5%). Men who have sex with men (MSM) (37.0%) was the predominant mode of HIV transmission, followed by heterosexual contact (35.4%) and injecting drug use (IDU) (21.6%). Austria was the nationality of most of the patients (63.8%); more than one third of individuals originated from high and low prevalence countries (16.6% and 19.6%, respectively). Half of the patients lived in a metropolitan area and about one third in an area counting less than 100,000 inhabitants. Almost half of the patients had a CD4 count more than three months after HIV diagnosis or had no CD4 count at all. About 15% of the patients had a CD4 count <200 cells/mm^3^ at HIV diagnosis and 13% of the patients never received cART. The median duration of cART of individuals ever on cART was 80.3 months, 80.9% of them had been on cART for at least nine months. Compared to survivors, patients who died were older (mean age: 40.1 versus 35.5 years), had a lower last measured mean CD4 cell count (269.0 versus 604.2 cells/mm^3^) and CD4 nadir (153.8 versus 269.5 cells/mm^3^), were more frequently infected through injecting drug use (45.3 versus 16.7%) but less often through men who have sex with men (24.2 versus 39.7%), lived less frequently outside Vienna (39.6 versus 48.5%) and were less likely to have ever been on ART (74.4 versus 89.7%). Of those ever on ART, patients who died had a lower mean duration of ART (75.5 versus 104.1 months). During observation, 1192 (17.4%) patients died, 380 (31.9%) of them due to AIDS-related diseases, 124 (10.4%) from liver-related diseases (including HCC), 139 (11.7%) of the deaths were non-AIDS-defining malignancies, 110 (9.2%) died due to non-AIDS infections, 85 (7.1%) due to cardiovascular diseases and 354 (29.7%) due to other or unknown causes. The median time of 51 days from the last visit at the clinic to death was much longer in patients with drug abuse, compared to 16 days in patients who died from cardiovascular disease, 1.5 days in patients with liver-related deaths and six days in patients who died from non-AIDS malignancies.

### 3.2. Changing Rates of Deaths

Table 2 shows mortality rates in different calendar time periods (period 1: 1997–2000, period 2: 2001–2004, period 3: 2005–2008, period 4: 2009–2011 and period 5: 2012–2014).

In the first period 251 patients died, in the second period 246, in the third period 288, in the fourth period 198 and in the fifth period 209 individuals during 7115.0, 10266.9, 14114.1, 13304.7 and 14904.9 person-years of follow-up. All-cause mortality rates decreased from 3.49 d/100py (95% CI: 3.08–3.95) in period 1 to 2.43 d/100py (95% CI: 2.14–2.75) in period 2 to 2.03 d/100py (95% CI: 1.81–2.28) in period 3 to 1.47 d/100py (95% CI: 1.28–1.69) in period 4 and 1.40 d/100py (95% CI: 1.22–1.61) in period 5, respectively. AIDS-related causes of death decreased over time from 1.62 d/100py (95% CI: 1.35–1.94) in period 1 to 0.28 d/100py (95% CI: 0.20–0.37) in period 5. Death due to liver-related diseases and Non-AIDS infections declined (0.38 d/100py (95% CI: 0.26–0.55) in period 1 to 0.17 d/100py (95% CI: 0.11–0.25) in period 5 and 0.27 d/100py (95% CI: 0.17–0.42) in period 1 to 0.10 d/100py (95% CI: 0.06–0.17) in period 5, respectively), whereas cardiovascular diseases as cause of death remained stable (0.27 d/100py (95% CI: 0.17–0.42) in period 1, 0.10 d/100py (95% CI: 0.05–0.18) in period 2, 0.16 d/100py (95%CI: 0.10–0.24) in period 3, 0.09 d/100py (95% CI: 0.05–0.16) in period 4 and 0.14 d/100py (95% CI: 0.09–0.22) in period 5, respectively) and deaths due to non-AIDS-defining malignancies increased (0.17 d/100py (95% CI: 0.10–0.30) in period 1 to 0.21 d/100py (95% CI: 0.15–0.30) in period 5). Mortality rates increased with age and decreased with CD4 count at HIV diagnosis as well as at initiation of cART in all five periods.

### 3.3. Predictors of All-Cause Mortality over Time

In adjusted Cox regression models higher age (compared to <30 years), IDU as HIV transmission category (compared to MSM) and lower CD4 counts (compared to a CD4 count of 500 cells/mm^3^) were significant factors associated with an increased mortality risk (Table 3) and these factors remained stable over all five periods.

A CD4 count of 50 cells/mm^3^, 200 cells/mm^3^ and 350 was associated with a higher risk of death compared to a CD4 count of 500 cells/mm^3^. In periods 2 to 5 a population size of residence area of less than 100,000 demonstrated a trend to a higher survival benefit compared to a population size of more than 1 million. A longer duration of cART decreased mortality risk in all five periods. In periods 3 to 5 the adjusted HR was higher in patients originating from Austria or other low prevalence countries compared to individuals from high prevalence countries.

### 3.4. Association with Specific Causes of Death

Multivariable Cox proportional hazard models including all time periods were calculated and controlled for time-updated CD4, age and cART, HIV transmission category, population size of area of residence and country of birth. Associations of latest CD4 counts with all-cause mortality and specific causes of death are shown in Figure 1.

The lower the CD4 count compared to latest CD4 counts of 500 cells/mm^3^ the higher the risk of dying applied to all causes of death except cardiovascular mortality, where no significant association could be observed.

### 3.5. Sensitivity Analyses

Results were similar in sensitivity analyses where observation time was divided into two periods: 1997–2004 and 2005–2014 (see Appendix A).

## 4. Discussion

Since the introduction of cART in our cohort we observed a marked and continuous decline in all-cause mortality rates. Death caused by AIDS-related diseases decreased most, and also liver-related diseases and non-AIDS infections decreased. However, cardiovascular diseases as cause of death remained stable and deaths due to non-AIDS-defining malignancies increased over time. This growing proportion of non-AIDS-defining illnesses as causes of death could be shown in different studies [15,17,21,22,23,24,25,26]. In the most recent period AIDS-defining diseases remained the most common cause of death; however, the difference with non-AIDS-defining malignancies, liver related diseases, non-AIDS-defining diseases and cardiovascular diseases was much smaller than in earlier periods. These findings are consistent with findings from other studies [15,26]. 

We found that individuals with latest CD4 counts below 500 cells/mm^3^ were more likely to die in all five observation periods compared to individuals with the latest CD4 counts of 500 cells/mm^3^. Lower CD4 counts refer to a higher risk of death caused by AIDS-related diseases, liver-related diseases, non-AIDS infections and non-AIDS-defining malignancies. No significant association was observed for cardiovascular mortality. Results were similar in sensitivity analyses where observation time was divided into two periods: 1997–2004 and 2005–2014. It indicated that a large proportion of HIV-positive individuals had progressed to late stages of infection before they were diagnosed and could be treated [27]. A recent study found that late presenters had poorer survival compared to non-late presenters in all time periods [28]. Half of all newly diagnosed individuals were late presenters in our cohort, defined as patients with a CD4 count <350 cells/μL at time of HIV diagnosis [29]. We also found that duration of cART was significantly associated with a reduced mortality risk. Moreover, increasing CD4 counts and age at death could be found over time [17], which is consistent with findings in our study. It is shown that patients dying from causes other than AIDS initiated cART at higher CD4 counts, had more experience in cART and were also rather treated near the time of death compared to individuals with AIDS-defining deaths [17]. The pronounced longer median time from last visit at the clinic to death in patients who died from drug abuse may be interpreted as proxy for being lost to follow-up or, more generally, as less adherent to cART.

Several studies investigated the association between immunodeficiency and non-AIDS-related deaths; however, most of them showed the implication of the latest CD4 count or CD4 cell measurements shortly before death [15,17,30]. A large cohort study showed a lower CD4 count at the start of ART to be associated with death due to AIDS, renal failure, AIDS infection, non-AIDS malignancy and other causes, but not with death due to AIDS malignancies. Furthermore, they found an association between higher CD4 counts at the start of ART and higher mortality rates due to violent causes [13]. A recent South African study found associations between lower CD4 cell counts and an increased risk of Kaposi sarcoma, cervical cancer, non-Hodgkin lymphoma and Hodgkin lymphoma [31]. Moreover, a large US study reported that Kaposi sarcoma and lymphoma rates were highest in the first six months of treatment particularly among individuals with low CD4 counts [32]. In our study we did not look at a distinct CD4 value and its effect on mortality but we considered all measured CD4 counts per patient during the entire observation period to evaluate the impact on all-cause mortality and specific causes of death [16,25].

One of our most striking analysis results showed lower CD4 cell counts not to be associated with death due to cardiovascular diseases. Similar findings were described in a large study which could not find any association between CD4 cell markers and a higher risk of cardiovascular disease death [22]. Moreover, a recent study reported higher adjusted HRs at 10 years after starting ART in individuals with CD4 values <200 cells/µL (compared to ≥200 cells/µL) for AIDS-, non-AIDS- and liver-related causes of death, but no significant association could be found for cardiovascular disease mortality [33]. One might also expect age-adjusted cardiovascular disease mortality to have decreased during the study period along with changes in cardiovascular disease risk factor management, thus obfuscating a potential association with a CD4 count. An age-adjusted decline of cardiovascular diseases has been observed in high income countries with access to healthcare [34].

In AHIVCOS, about 15% of patients were also positive for hepatitis C virus (HCV). These patients have an increased risk of morbidity and mortality compared with individuals with HIV infection alone [35]. Qurishi et al. showed an improved overall survival in HCV-coinfected individuals treated with HAART and a reduced liver-related mortality risk [36]. It was also found that patients who died of liver-related causes had a higher median CD4 nadir and latest CD4 counts compared with individuals dying of AIDS-defining diseases [37]. Our observation strengthens these findings, in that a higher percentage of patients who died of HBV/HCV-associated causes had a CD4 count of more than 200 cells/mm^3^ at HIV diagnosis as well as at initiation of cART, compared to patients who died from AIDS-related conditions. Moreover, in HCV-coinfected patients time between HIV diagnosis and initiation of cART was much higher compared to all other causes of death.

In our study, we found that IDU had higher mortality rates and a higher risk of death in all calendar time periods compared to individuals who acquired HIV through other transmission categories. It has been shown that patients infected through IDU have less access to cART [38]. Adherence to cART is important as treatment interruptions are associated with poor outcomes [39] and development of resistances [40]. However, there are controversial findings concerning adherence to cART in injecting drug users. Studies indicated a poorer response to cART as a result of poor adherence; hence the benefits of cART appear to be a less strong contributing factor to a higher mortality risk [41,42], whereas other data could not confirm this finding [38].

One of the strengths of this study is its design: It is an open, observational study with complete long-term follow-up. AHIVCOS provides a good representation of HIV-positive individuals in Austria; therefore, a selection bias could be minimized. Causes of death were standardized classified according to the CoDe system [20] which implies differentiation between immediate, contributing and underlying causes of death. All deaths were reviewed centrally by one physician, thus minimizing information bias possibly resulting from different methods and/or clinicians when determining a cause of death. If individuals are lost to follow-up their vital status is proved by cross-checking with data from the National Death Registry.

Despite the strength of this study, it is worth mentioning its limitations too. We could not adjust for socioeconomic and lifestyle factors like smoking, as data were missing or incomplete. Moreover, the relationship between health seeking behaviour and lifestyle or socioeconomic factors and their impact on late presentation is not known. As a result of loss to follow-up of patients who left the country, particularly migrants, mortality rates in these groups may be underestimated. Furthermore, we did not distinguish between myocardial infarction type 1 and type 2 concerning cardiovascular mortality.

## 5. Conclusions

Since the introduction of cART, risk of death continuously decreased and causes of death changed. In the present study, an association of lower CD4 counts with increased mortality for most causes of death was found. However, we do not find evidence of a higher cardiovascular mortality risk in individuals with a low CD4 count.

## Figures and Tables

**Figure 1 ijerph-18-12532-f001:**
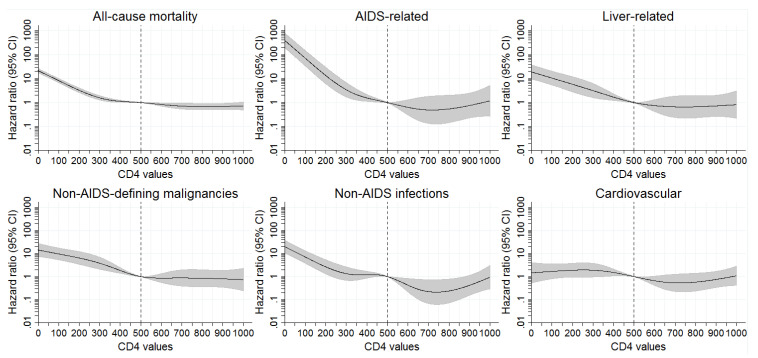
Associations of latest CD4 counts with all-cause mortality and specific causes of death. Results from adjusted Cox regression models including all time periods.

**Table 1 ijerph-18-12532-t001:** Characteristics of patients.

	Total		Survivors		Non-Survivors		*p*-Value *
N	6848	(100.0)	5656	(100.0)	1192	(100.0)	
Sex							0.015
Male	5100	(74.5)	4179	(73.9)	921	(77.3)	
Female	1748	(25.5)	1477	(26.1)	271	(22.7)	
Age at study entry (years)	36.3 ± 11.2		35.5 ± 10.8		40.1 ± 12.4		<0.001
	34.5 (28.5–42.5)		34.1 (27.9–41.5)		37.5 (32.1–46.7)		
Age group at study entry (years)							<0.001
<30	2139	(31.2)	1902	(33.6)	237	(19.9)	
30–39	2574	(37.6)	2108	(37.3)	466	(39.1)	
40–49	1349	(19.7)	1097	(19.4)	252	(21.1)	
≥50	786	(11.5)	549	(9.7)	237	(19.9)	
HIV transmission category							<0.001
Male injecting drug user	1030	(15.0)	639	(11.3)	391	(32.8)	
Female injecting drug user	453	(6.6)	304	(5.4)	149	(12.5)	
Male heterosexual	1216	(17.8)	1049	(18.6)	167	(14.0)	
Female heterosexual	1203	(17.6)	1100	(19.5)	103	(8.6)	
Men who have sex with men	2533	(37.0)	2245	(39.7)	288	(24.2)	
Other	413	(6.0)	319	(5.6)	94	(7.9)	
CD4 Nadir (cells/mm^3^)	250.3 ± 200.1		269.5 ± 196.5		153.8 ± 189.7		<0.001
	220.0 (93.0–354.0)		244.0 (122.0–374.0)		96.0 (24.0–202.0)		
Last CD4 count (cells/mm^3^)	548.7 ± 320.1		604.2 ± 300.0		269.0 ± 267.5		<0.001
	527.0 (321.0–736.0)		578.0 (399.0–774.0)		196.0 (62.0–396.0)		
CD4 count at HIV diagnosis (cells/mm^3^)							<0.001
<50	428	(6.3)	322	(5.7)	106	(8.9)	
50–199	587	(8.6)	496	(8.8)	91	(7.6)	
200–349	773	(11.3)	711	(12.6)	62	(5.2)	
≥350	1728	(25.2)	1601	(28.3)	127	(10.7)	
No CD4 count within 3 months	3332	(48.7)	2526	(44.7)	806	(67.6)	
CD4 count at initiation of cART (cells/mm^3^)							<0.001
<50	601	(8.8)	472	(8.4)	129	(10.8)	
50–199	1090	(15.9)	875	(15.5)	215	(18.0)	
200–349	1368	(20.0)	1254	(22.2)	114	(9.6)	
≥350	1273	(18.6)	1196	(21.2)	77	(6.5)	
No CD4 count within 3 months	2516	(36.7)	1859	(32.9)	657	(55.1)	
Ever cART							
Duration of cART (months)	99.8 ± 76.0		104.1 ± 77.2		75.5 ± 63.4		<0.001
	80.3 (36.7–155.6)		84.6 (39.5–165.8)		60.0 (23.3–112.8)		
Duration of cART							<0.001
<9 months	420	(6.1)	288	(5.1)	132	(11.1)	
9–18 months	368	(5.4)	310	(5.5)	58	(4.9)	
>18 months	5173	(75.5)	4477	(79.2)	696	(58.4)	
Never cART	887	(13.0)	581	(10.3)	306	(25.7)	
Population size of area of residence							<0.001
<100,000	2259	(33.0)	1964	(34.7)	295	(24.8)	
≥100,000	958	(14.0)	782	(13.8)	176	(14.8)	
>1 million	3552	(51.9)	2862	(50.6)	690	(57.9)	
Missing value	79	(1.2)	48	(0.9)	31	(2.6)	
Country of birth							<0.001
Austria	4369	(63.8)	3499	(61.9)	870	(73.0)	
Other low prevalence countries	1340	(19.6)	1213	(21.5)	127	(10.7)	
High prevalence countries	1139	(16.6)	944	(16.7)	195	(16.4)	
Follow-up time (years)	8.72 ± 5.77		9.35 ± 5.78		5.73 ± 4.69		<0.001
	7.78 (3.75–13.67)		8.64 (4.35–15.00)		4.63 (1.70–8.83)		
Deaths					1192	(100.0)	
Causes of death							
AIDS-related					380	(31.9)	
Liver-related ^1^					124	(10.4)	
Non-AIDS-defining malignancies					139	(11.7)	
Non-AIDS infections					110	(9.2)	
Cardiovascular					85	(7.1)	
Drug abuse					120	(10.1)	
Suicide					31	(2.6)	
Other ^2^					132	(11.1)	
Unknown					71	(6.0)	

Values are presented as mean ± SD and median (interquartile range (IQR)) or number (%). * Comparison between survivors and non-survivors (χ^2^ test or Mann-Whitney U test). ^1^ Liver-related including HCV with cirrhosis (N = 76), HCV with liver failure (N = 15), HCV with liver cancer (N = 4), HCV without categorization (N = 11), HBV with cirrhosis (N = 10), HBV with liver failure (N = 1), HBV with liver cancer (N = 7). ^2^ Other causes: Accident or violent death (N = 39), chronic obstructive lung disease (N = 25), renal failure (N = 12), liver failure (other than HBV/HCV) (N = 5), lung embolus (N = 5), pancreatitis (N = 5), CNS disease (N = 2), gastro-intestinal haemorrhage (N = 2), bleeding (haemophilia) (N = 1), diabetes mellitus (N = 1), digestive system disease (N = 1), lactic acidosis (N = 1), primary pulmonary hypertension (N = 1), unclassifiable causes (N = 3) and others (N = 29).Values are presented as mean ± SD and median (interquartile range (IQR)) or number (%).

**Table 2 ijerph-18-12532-t002:** Mortality rates in different calendar time periods.

Observation Periods	1997–2000	2001–2004	2005–2008	2009–2011	2012–2014
Number of deaths	251	246	288	198	209
Person-years of follow-up	7115.0	10,266.9	14,114.1	13,304.7	14,904.9
Mortality rate per 100 person-years	3.49 (3.08–3.95)	2.43 (2.14–2.75)	2.03 (1.81–2.28)	1.47 (1.28–1.69)	1.40 (1.22–1.61)
Causes of death					
AIDS-related	1.62 (1.35–1.94)	0.88 (0.72–1.08)	0.62 (0.51–0.77)	0.34 (0.25–0.45)	0.28 (0.20–0.37)
Liver-related	0.38 (0.26–0.55)	0.28 (0.20–0.41)	0.18 (0.13–0.27)	0.13 (0.08–0.21)	0.17 (0.11–0.25)
Non-AIDS-defining malignancies	0.17 (0.10–0.30)	0.21 (0.14–0.33)	0.26 (0.18–0.35)	0.29 (0.21–0.39)	0.21 (0.15–0.30)
Non-AIDS infections	0.27 (0.17–0.42)	0.27 (0.19–0.39)	0.21 (0.14–0.30)	0.14 (0.09–0.22)	0.10 (0.06–0.17)
Cardiovascular	0.27 (0.17–0.42)	0.10 (0.05–0.18)	0.16 (0.10–0.24)	0.09 (0.05–0.16)	0.14 (0.09–0.22)
Drug abuse ^1^	0.17 (0.10–0.30)	0.27 (0.19–0.39)	0.23 (0.16–0.32)	0.17 (0.11–0.25)	0.17 (0.12–0.26)
Suicide	0.07 (0.03–0.17)	0.13 (0.07–0.22)	0.04 (0.02–0.09)	0.02 (0.00–0.06)	0.03 (0.01–0.08)
Other ^2^	0.32 (0.21–0.49)	0.16 (0.10–0.25)	0.27 (0.20–0.37)	0.18 (0.12–0.27)	0.20 (0.14–0.29)
Unknown	0.22 (0.14–0.37)	0.13 (0.07–0.22)	0.07 (0.04–0.13)	0.13 (0.08–0.21)	0.10 (0.06–0.17)
Age at start of each period (years)					
<30	1.66 (1.17–2.36)	1.84 (1.35–2.50)	0.99 (0.68–1.43)	0.90 (0.57–1.41)	0.57 (0.32–1.03)
30–39	3.58 (2.99–4.28)	2.01 (1.63–2.48)	1.71 (1.38–2.12)	1.02 (0.75–1.39)	1.03 (0.76–1.40)
40–49	4.15 (3.18–5.42)	2.89 (2.29–3.64)	2.14 (1.75–2.62)	1.58 (1.26–2.00)	1.22 (0.95–1.57)
≥50	7.12 (5.30–9.57)	4.00 (3.03–5.30)	4.01 (3.23–4.97)	2.42 (1.89–3.08)	2.44 (1.99–2.99)
HIV transmission category					
Male injecting drug user	6.43 (5.23–7.92)	5.02 (4.08–6.19)	4.31 (3.52–5.28)	3.63 (2.83–4.64)	3.28 (2.53–4.25)
Female injecting drug user	4.06 (2.89–5.71)	3.02 (2.11–4.32)	3.08 (2.21–4.28)	2.51 (1.65–3.81)	3.20 (2.22–4.60)
Male heterosexual	2.29 (1.49–3.51)	1.67 (1.15–2.44)	1.83 (1.37–2.46)	1.72 (1.26–2.33)	1.21 (0.85–1.71)
Female heterosexual	1.44 (0.88–2.35)	1.40 (0.94–2.06)	0.82 (0.55–1.24)	0.79 (0.52–1.21)	0.59 (0.37–0.95)
Other	4.81 (3.23–7.18)	4.36 (3.01–6.32)	1.59 (0.93–2.75)	1.33 (0.72–2.48)	2.37 (1.51–3.72)
Men who have sex with men	2.72 (2.13–3.46)	1.47 (1.11–1.93)	1.63 (1.30–2.03)	0.80 (0.58–1.09)	0.93 (0.71–1.21)
CD4 count at HIV diagnosis (cells/mm^3^)					
<50	11.19 (7.50–16.70)	5.04 (3.41–7.47)	4.14 (2.94–5.82)	2.34 (1.49–3.66)	2.82 (1.93–4.11)
50–199	7.29 (5.24–10.16)	2.55 (1.64–3.95)	2.58 (1.84–3.63)	1.14 (0.68–1.88)	1.73 (1.18–2.52)
200–349	1.85 (1.00–3.44)	2.11 (1.39–3.20)	1.62 (1.11–2.36)	1.10 (0.70–1.72)	0.75 (0.46–1.23)
≥350	2.22 (1.54–3.22)	1.84 (1.36–2.50)	1.33 (1.01–1.76)	1.12 (0.83–1.50)	0.87 (0.64–1.17)
CD4 count at initiation of cART (cells/mm^3^)					
<50	5.53 (4.04–7.57)	3.68 (2.66–5.07)	3.01 (2.21–4.09)	1.69 (1.10–2.60)	2.11 (1.47–3.04)
50–199	2.47 (1.83–3.34)	2.09 (1.59–2.76)	2.47 (1.97–3.08)	2.20 (1.70–2.84)	2.45 (1.93–3.10)
200–349	1.04 (0.64–1.69)	1.23 (0.87–1.75)	1.15 (0.86–1.56)	1.22 (0.90–1.64)	1.06 (0.78–1.43)
≥350	0.93 (0.54–1.61)	1.28 (0.87–1.88)	1.20 (0.87–1.67)	0.58 (0.37–0.91)	0.72 (0.50–1.03)
Population size of residence area					
<100,000	2.82 (2.19–3.64)	2.02 (1.57–2.59)	1.56 (1.24–1.97)	0.83 (0.60–1.14)	1.19 (0.93–1.53)
≥100,000	3.89 (2.85–5.30)	2.30 (1.59–3.33)	1.85 (1.31–2.60)	2.06 (1.51–2.82)	1.67 (1.20–2.33)
>1 million	3.36 (2.83–3.98)	2.65 (2.27–3.10)	2.29 (1.97–2.65)	1.69 (1.40–2.03)	1.49 (1.24–1.79)
Country of birth					
Austria	2.77 (2.35–3.26)	2.45 (2.11–2.83)	2.28 (2.00–2.61)	1.74 (1.49–2.04)	1.68 (1.44–1.95)
Other low prevalence countries	0.84 (0.42–1.68)	1.74 (1.20–2.52)	1.62 (1.19–2.22)	1.03 (0.70–1.53)	0.98 (0.67–1.42)
High prevalence countries	9.96 (8.15–12.16)	3.16 (2.32–4.29)	1.38 (0.97–1.98)	0.78 (0.47–1.29)	0.60 (0.34–1.05)

Mortality rate (deaths/100 person-years) with 95% confidence intervals. ^1^ Drug abuse including chronic alcohol abuse (N = 21), chronic intravenous drug-use (N = 38) as well as acute intoxication (N = 34) and drug abuse without categorization (N = 27). ^2^ Other causes: Accident or violent death (N = 39), chronic obstructive lung disease (N = 25), renal failure (N = 12), liver failure (other than HBV/HCV) (N = 5), lung embolus (N = 5), pancreatitis (N = 5), CNS disease (N = 2), gastro-intestinal haemorrhage (N = 2), bleeding (haemophilia) (N = 1), diabetes mellitus (N = 1), digestive system disease (N = 1), lactic acidosis (N = 1), primary pulmonary hypertension (N = 1), unclassifiable causes (N = 3) and others (N = 29).

**Table 3 ijerph-18-12532-t003:** Factors associated with all-cause mortality in different calendar time periods: Results from multivariable Cox regression models.

Observation Periods	All-Cause Mortality									
	1997–2000		2001–2004		2005–2008		2009–2011		2012–2014	
	aHR	(95% CI)	aHR	(95% CI)	aHR	(95% CI)	aHR	(95% CI)	aHR	(95% CI)
Age (years) (time-updated)										
≥50	4.29	(2.28–8.10)	3.43	(2.02–5.81)	4.37	(2.54–7.51)	2.43	(1.35–4.39)	3.77	(1.80–7.88)
40–49	2.08	(1.16–3.72)	1.59	(0.98–2.59)	1.89	(1.12–3.18)	1.45	(0.82–2.57)	1.99	(0.94–4.23)
30–39	1.50	(0.85–2.63)	1.06	(0.66–1.70)	1.45	(0.85–2.47)	0.83	(0.44–1.53)	1.49	(0.69–3.25)
<30	1.00	(Ref.)	1.00	(Ref.)	1.00	(Ref.)	1.00	(Ref.)	1.00	(Ref.)
HIV transmission category										
Male injecting drug user	2.61	(1.72–3.96)	3.50	(2.36–5.19)	2.05	(1.46–2.88)	2.24	(1.43–3.49)	2.37	(1.58–3.55)
Female injecting drug user	1.80	(1.06–3.06)	2.55	(1.51–4.31)	2.21	(1.43–3.43)	1.64	(0.91–2.93)	2.56	(1.57–4.18)
Male heterosexual	0.72	(0.40–1.31)	0.75	(0.45–1.26)	0.83	(0.56–1.25)	1.80	(1.13–2.85)	1.02	(0.63–1.63)
Female heterosexual	0.56	(0.29–1.08)	0.93	(0.54–1.62)	0.67	(0.41–1.10)	1.01	(0.56–1.82)	0.68	(0.37–1.25)
Other	1.93	(1.02–3.66)	2.91	(1.71–4.94)	0.93	(0.49–1.76)	1.58	(0.75–3.31)	2.28	(1.31–3.98)
Men who have sex with men	1.00	(Ref.)	1.00	(Ref.)	1.00	(Ref.)	1.00	(Ref.)	1.00	(Ref.)
CD4 count at specific values (cells/mm^3^) *										
50	19.50	(11.50–33.06)	13.10	(8.86–19.38)	17.26	(11.68–25.51)	21.69	(12.86–36.60)	13.42	(8.60–20.95)
200	4.05	(2.27–7.23)	2.65	(1.65–4.23)	3.59	(2.28–5.64)	5.19	(2.98–9.02)	4.00	(2.71–5.90)
350	2.06	(1.19–3.59)	1.14	(0.87–1.50)	1.31	(0.96–1.78)	1.61	(1.00–2.61)	1.44	(0.99–2.11)
500	1.00	(Ref.)	1.00	(Ref.)	1.00	(Ref.)	1.00	(Ref.)	1.00	(Ref.)
cART (time-updated)	0.49	(0.34–0.72)	0.52	(0.38–0.71)	0.87	(0.65–1.18)	0.54	(0.37–0.80)	0.73	(0.48–1.11)
Population size of residence area										
<100,000	1.14	(0.78–1.66)	0.77	(0.55–1.07)	0.84	(0.62–1.13)	0.50	(0.33–0.75)	0.84	(0.59–1.18)
≥100,000	1.00	(0.64–1.56)	0.74	(0.47–1.16)	0.84	(0.57–1.24)	1.29	(0.87–1.91)	1.25	(0.83–1.88)
>1 million	1.00	(Ref.)	1.00	(Ref.)	1.00	(Ref.)	1.00	(Ref.)	1.00	(Ref.)
Country of birth										
Austria	0.28	(0.20–0.40)	0.71	(0.45–1.14)	1.17	(0.75–1.81)	2.36	(1.16–4.80)	1.85	(0.95–3.61)
Other low prevalence countries	0.18	(0.09–0.39)	0.77	(0.44–1.37)	1.13	(0.67–1.92)	1.80	(0.82–3.94)	1.56	(0.73–3.31)
High prevalence countries	1.00	(Ref.)	1.00	(Ref.)	1.00	(Ref.)	1.00	(Ref.)	1.00	(Ref.)

* The adjusted hazard ratios represent a comparison of the point estimates of a CD4 count of 50, 200 and 350 cells/mm^3^ versus a CD4 count of 500 cells/mm^3^ extracted from the model. aHR, adjusted hazard ratio.

## Data Availability

The data underlying this Article will be shared on reasonable request to the corresponding author, subject to approval by the AHIVCOS cohort scientific committee.

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
