# Peer review of "A Lower CD4 Count Predicts Most Causes of Death except Cardiovascular Deaths. The Austrian HIV Cohort Study"

_ijerph, 2021, doi:10.3390/ijerph182312532_

Round 1

Reviewer 1 Report

The present manuscript investigates the evolving causes of death in HIV-infected subjects under cART in a big open Austrian cohort.  The finding are rather confirmatory of previous publications and some important variables such as smoking and lifestyle are missing (as the authors admit).  However, the study has been performed in a very meticulous way; findings are very sound and interesting to the HIV clinicians community.  

There is some lack of clarity with regard to “liver-related causes of death”.  It would be good to define this concept in the “Classification of causes of death” paragraph and briefly summarize it underneath Table 1.  Are all subjects in this category HBV/HCV positive or are other causes of liver cirrhosis (e.g. alcohol abuse) also part of this category?  What is the overlap with intravenous drug abuse?  Can both be disentangled in a multivariate analysis?  These questions should be clarified in the Results section and will also modify the Discussion (lines 285-304).

In the Introduction line 58, two studies (ref 5,6) are mentioned, showing that “viral load was hardly predictive of death”.   These papers date back to 2004 and 2006, an era when HAART was very complicated and difficult to strictly adhere to. The present study covers the same period, but also a more recent period, when cART is more patient-friendly, with possible implications for viral control.  I would like to see some analysis regarding the potential effect of viral suppression on mortality e.g.  comparing patients with undetectable load (with possible exception of blips < 1000 copies) and those with poorly controlled viral load despite cART treatment.

Reviewer 2 Report

This is an interesting and well-constructed analysis of observational data from the Austrian HIV Cohort Study between 1997 and 2014, which provides strong evidence of stable rates of cardiovascular mortality over the study period. This is set against decreasing rates of other HIV-associated conditions including AIDS- and non-AIDS-associated infections and liver disease-related mortality. The evidence that baseline and time-updated CD4 T cell counts are not associated with cardiovascular mortality risk appears to be appropriate and rigorous. The manuscript is well written and appropriately acknowledges relevant limitations.

It is interesting to note that cardiovascular mortality rates remained statistically stable over the 17 year period of observation, which argues that duration of HIV treatment was not a significant risk factor. It would be useful to include this analysis in relation to the discussion of CVD mortality in this study. In this regard, previous studies (D:A:D most notably) have demonstrated increased CVD mortality risk associated with specific treatment regimens (eg. duration of indinavir or darunavir, current use of abacavir) so it would be interesting to note the use of these medications within the cohort.

It would also be informative to compare the underlying CVD mortality rate in this cohort (~0.1-0.2/100py) with data from the background population if this is available (eg. J Am Coll Cardiol. 2020 Dec 22;76(25):2982-3021), given the expectation that age-adjusted CVD mortality would decrease during this study period with changes in CVD risk factor management.

Round 2

Reviewer 1 Report

Extensive and sufficient clarification on most remarks.  Only the last question on viral suppression has not been addressed.